# Dietary Patterns and the Double Burden of Malnutrition in Mexican Adolescents: Results from ENSANUT-2006

**DOI:** 10.3390/nu11112753

**Published:** 2019-11-13

**Authors:** Arli Guadalupe Zárate-Ortiz, Alida Melse-Boonstra, Sonia Rodríguez-Ramírez, Sonia Hernández-Cordero, Edith J. M. Feskens

**Affiliations:** 1Division of Human Nutrition and Health, Wageningen University & Research, Wageningen 6708WE, The Netherlands; arli.zarateortiz@wur.nl (A.G.Z.-O.);; 2Research Center of Nutrition and Health, National Institute for Public Health, Cuernavaca 62100, Mexico; 3Universidad Iberoamericana, Mexico City 01219, Mexico

**Keywords:** dietary patterns, double burden, adolescence, malnutrition, overweight and obesity, anemia, stunting

## Abstract

Mexico is facing the double burden of malnutrition, and adolescents are not an exception. Diet plays an important role, both in causing overweight and undernutrition. This study aimed to describe the dietary patterns (DPs) of Mexican adolescents and to examine its association with nutritional status using data from adolescents aged 12–19 years (*n* = 7380) from the National Survey of Health and Nutrition (ENSANUT-2006). Principal component analysis was used to derivate the DPs. Associations between DP and nutritional status were determined by prevalence ratio (PR). Four DPs were identified: nontraditional and breakfast-type, Western, plant-based, and protein-rich. The prevalence of overweight and obesity was higher in adolescents who scored high on the Western pattern (PR: 1.15, 95% CI: 1.08–1.21) or on the plant-based pattern (PR: 1.09, 95% CI: 1.03–1.17). The Western pattern was positively associated with anemia in girls (PR: 1.18, 95% CI: 1.03–1.35), while the nontraditional and breakfast-type pattern was inversely associated with anemia in adolescents aged 12–15 years (PR: 0.87, 95% CI: 0.76–0.99) and in girls (PR: 0.84, 95% CI: 0.75–0.97). The Western and plant-based patterns were simultaneously associated with overweight–obesity and at least one indicator of undernutrition. In the context of the double burden of malnutrition, dietary advice must consider malnutrition in all its forms.

## 1. Introduction

After childhood, adolescence is the second important developmental period in the life course [1]. Complex interactions of physiological, sexual, neurological, and behavioral factors prompt the transition from childhood to adult life [1,2]. Adolescence is a period characterized by rapid growth and changes in body composition, leading to increased energy and micronutrient requirements [1].

Eating behavior of adolescents may be influenced by sociodemographic characteristics, such as socioeconomic level, education level, ethnicity, and gender [3]. Dietary habits established during adolescence tend to persist during adulthood. Moreover, dietary patterns adopted during this stage may contribute to health outcomes later in life [1,4]. Failure to comply with nutritional demands during adolescence can lead to growth retardation, impaired organ remodeling, and micronutrient deficiencies [1]. In addition, evidence suggests that, in adolescents, a Western-type dietary pattern (i.e., an industrialized diet, mainly characterized by high energy, fat, and sugar content) is associated with the prevalence of obesity [5,6], metabolic risk [7], and depression [8,9].

The double burden of malnutrition is characterized by the coexistence of micronutrient deficiencies and diet-related chronic conditions. This problem can be present at individual, household, and/or national levels [10]. In Mexico, micronutrient deficiency, anemia, and stunting are still prevalent among adolescents [11,12], while the prevalence of overweight and obesity has increased dramatically during the last 13 years. In 2012, more than 30% of Mexican adolescents were overweight or obese [13]. Furthermore, other metabolic disorders related to poor dietary quality, such as type 2 diabetes and metabolic syndrome, have started to emerge in this age group [14,15].

Up until now, adolescents’ dietary intake at a national scale in Mexico has only been examined in terms of glycemic index and intake of single nutrients and food group consumption [16,17,18]. However, the single nutrient approach has some limitations when the aim is to demonstrate associations between diet and health outcomes. People do not eat isolated foods or nutrients but rather consume meals consisting of a variety of foods and nutrients [5]. Dietary patterns present a broader picture of a combination of foods and nutrients, including the antagonist, additive, and synergetic effect of the food matrix [19]. Thus, the dietary pattern approach is a better alternative to study the interactions between diet and health. Therefore, the primary aim of this study was to describe dietary patterns of Mexican adolescents and to associate these with overweight–obesity, anemia, and stunting.

## 2. Materials and Methods

### 2.1. Study Design and Study Population

This study used data from the National Health and Nutrition Survey (ENSANUT-2006). ENSANUT-2006 is a probabilistic, cross-sectional, stratified cluster sample study representative at national, regional, and state levels and that of urban and rural areas in Mexico. The survey was conducted from October 2005 to May 2006. The description of sampling procedures is detailed in Palma et al. [20]. ENSANUT-2006 aimed to characterize the health and nutritional status of different age groups in the Mexican population. For the purposes of this study, we used the adolescents’ subsample, which includes 8768 boys and girls aged 12 to 19 years. A flow chart of the selection process of the study population is shown in Figure 1. We excluded 69 adolescent girls that were pregnant, lactating, or both because nutritional requirements and body composition change during these stages. Secondly, we excluded subjects with missing or implausible values for anthropometric information (*n* = 549) and with implausible reported energy intake (*n* = 549). For exclusion criteria, we considered a plausible body mass index (BMI) to be > 10 and < 58 kg/m^2^ [13]. In addition, we used the energy intake/basal metabolic rate ratio (EI:BMR) cutoffs > 0.5 and below 3 standard deviations (SD) to assess plausible reported energy intake [21,22]. After exclusions, 7380 subjects with detailed information of diet and anthropometry remained, representing around 7,078,490 adolescents. Out of the 7380 adolescents, a subsample of 7080 (representing 6,868,671 adolescents) had data on hemoglobin concentrations. To facilitate the interpretation of the results, we classified the study subjects into two adolescent stages: 12–15 and 16–19 years.

### 2.2. Population Characteristics

Information on several sociodemographic characteristics was obtained. The country was divided into three geographic regions that have similar characteristics: north, center, and south. The living area was defined as the type of locality, i.e., (1) urban for a population of ≥ 2500 inhabitants and (2) rural for a population of < 2500 inhabitants. Socioeconomic level was classified into tertiles: (1) low, (2) medium, and (3) high. The classification was based on household characteristics, goods, and availability of basic services [20].

For nutritional status data, staff with standardized training measured the weight and height; details of the anthropometric techniques and materials are described elsewhere [23]. The indicators height-for-age Z-score (HAZ) and BMI-for-age Z-score (BAZ) were used to classify nutritional status according to the World Health Organization (WHO) standards [24]. Stunting was defined as HAZ below 2 SD, and overweight/obesity was defined as BAZ above 1 SD. Hemoglobin concentrations were adjusted for residential altitude [25]. Anemia was defined according to the WHO cutoff points per gender and age group as follows: boys aged 12–14 years and all girls, hemoglobin concentrations <120 g/L; boys aged >15 years, hemoglobin concentrations <130 g/L [25].

### 2.3. Dietary Data

Trained staff administered a seven-day food frequency questionnaire (FFQ). This FFQ includes 101 food items and is based on a previously validated FFQ [26]. For the current analysis, we classified the 101 food items into 30 food groups according to their nutritional profile and cultural relevance. For deriving the dietary patterns, we excluded those food groups consumed by less than 10% of the adolescents (alcoholic drinks, drinks without energy, and nutritional supplements). Appendix A displays examples of each food group and the percentage of consumers.

To identify the dietary patterns, principal component analysis (PCA) was used. We estimated mean intake per food group (g/day) for each subject and standardized the values with Z-scores. Then, the 26 food groups were consolidated into principal components and rotated orthogonally (varimax rotation) to facilitate interpretability. The number of patterns to be retained was chosen on the basis of the scree plot and eigenvalue > 1.0. Four dietary patterns were retained, which accounted for a cumulative variation of 28.0% in food intake. Food groups with factor loadings ≥ 0.30 were considered as key contributors to the dietary patterns and used to label them.

### 2.4. Statistical Analysis

All analyses were performed using IBM SPSS Statistics 22. PCA provided a factor score for each subject for each of the four dietary patterns. We categorized the factor scores of each dietary pattern into quartiles. Energy and nutrient intake across the four dietary patterns were calculated and compared. Population characteristics and prevalence ratios (PR) were calculated using CSPLAN for complex samples. Basic statistics were performed to describe the sociodemographic characteristics of our study population. In order to identify possible confounders, we compared sociodemographic characteristics (urban/rural, geographic region, and socioeconomic level) and nutritional status (BMI/age, anemia, and stunting) among adolescent age and gender groups using the *F*-statistic (corrected and weighted chi-square).

Next, we investigated the association between nutritional status (overweight–obesity, anemia, and stunting) and dietary patterns by Cox regression as it is considered an alternative for logistic regression in cross-sectional studies when the outcome of interest is not rare [27]. PR for overweight–obesity and stunting were calculated by adjusting for the sociodemographic characteristics. EI:BMR ratio and BAZ were inversely correlated, suggesting underreporting in obese subjects. Thus, we additionally adjusted for EI:BMR ratio in order to control for misreporting (PRs unadjusted for EI:BMR ratio are shown in Appendix A). The PR for anemia was adjusted for sociodemographic characteristics and BAZ because evidence suggests a positive association between BMI and iron deficiency [28]. Cox regression with constant survival time gives a good approximation of the standard error of the prevalence ratio when robust variance estimates are used [27], but because this procedure was not present in SPSS, we calculated the confidence intervals using the Wald test result from logistic regression of the same model and the beta coefficient from the Cox regression analysis. The analysis of the association between nutritional indicators and dietary patterns was also conducted separately for adolescents’ age and sex groups because major changes in body composition and growth spurt occur during early adolescence (10–15 years), and differences in nutritional requirements between girls and boys are notable during adolescence [1].

Because energy and nutrient intakes were not normally distributed, Kruskal–Wallis *H*-test was used to assess differences in energy and nutrient intake across the quartiles in each pattern. Subsequently, we used Spearman correlation coefficients between intake of energy, macro- and micronutrients, and each dietary pattern.

## 3. Results

Sociodemographic characteristics and nutritional status of Mexican adolescents from ENSANUT-2006 are described by sex and age groups in Table 1. The sample characteristics are comparable to the Mexican population characteristics reported in 2006 [13,17]. Anemia was significantly higher in the group of boys aged 12–15 years compared to boys aged 16–19 years. In contrast, for adolescent girls, weight status and anemia were not significantly different between age groups. From the total number of subjects with anemia, about 30% presented overweight or obesity simultaneously.

### 3.1. Dietary Patterns

Four dietary patterns were identified: (1) nontraditional and breakfast-type, (2) Western, (3) plant-based, (4) protein-rich. In total, the four patterns accounted for 28% of the variance in the dietary variables. Details of the mean intake (g/day) and factor loading for food groups in each dietary pattern are listed in Table 2, and the correlation between nutritional content and the dietary pattern scores is shown in Table 3. The nontraditional and breakfast-type pattern was characterized by a high intake of milk, breakfast cereals, dairy, sweets, and sandwiches and a low intake of tortilla (Table 2). The nontraditional and breakfast-type pattern was positively correlated with energy from protein and fat and inversely correlated with fiber and EI:BMR ratio (Table 3). This pattern accounted for 12.0% of the variance in the diet. The Western pattern was characterized by high intake of industrialized sweet drinks, salty snacks, charcuterie, saturated fat, sandwiches, fast-food, and cereals/tubers. The Western pattern showed a positive correlation with the intake of total energy as well as percentage of energy from fat and sugar and contributed to 6.3% of the variance in the diet. The plant-based pattern was represented by high consumption of fruit, nonindustrialized sweet drinks, vegetables, avocado and nuts, maize-based food, fried vegetarian dishes, and sweet bakery. The plant-based pattern was positively correlated with total energy and intake of micronutrients, such as calcium, iron, zinc, vitamin C, vitamin A, and folate, and aggregated 5.2% of the variance in the diet. The protein-rich pattern was represented by a high intake of legumes, pasta and rice, eggs, soup, and poultry and red meat. The protein-rich pattern was highly correlated with total percentage of energy from protein as well as intake of iron and zinc and accounted for 4.5% of the diet variance. The content of energy, macronutrients, and micronutrients across the quartiles of the four dietary patterns is presented in Appendix A.

### 3.2. Association of Dietary Patterns with Overweight and Obesity

The prevalence of overweight and obesity was higher in adolescents who scored high on the Western pattern (PR: 1.15, 95% CI: 1.08–1.21) and on the plant-based pattern (PR: 1.10, 95%: CI 1.03–1.17) after taking into account the potential confounders of age, sex, living area, socioeconomic status, and region (Table 4), indicating an increase of 15% and 10%, respectively, per quartile of dietary pattern score. This increase was more pronounced in the younger adolescents (12–15 years) and in boys. A positive association was also seen for the plant-based pattern in the younger adolescents and in girls. The positive association between nontraditional and breakfast-type pattern and overweight/obesity in younger adolescents depended on inclusion of EI:BMR ratio in the model; after adjusting for EI:BMR ratio, this association turned to negative and nonsignificant (see Table 4 and Appendix A). The age group 12–15 years and girls with high scores in the protein-rich pattern were less likely to be overweight–obese, but this association disappeared after adjustment for EI:BMR ratio.

### 3.3. Association of Dietary Patterns with Anemia

None of the four dietary patterns were independently associated with the prevalence of anemia in the total study population. However, girls who scored high on the Western pattern presented higher prevalence of anemia (PR: 1.18, 95% CI: 1.03–1.35) (Table 5). In contrast, the nontraditional and breakfast-type pattern was inversely associated with anemia in younger adolescents (12–15 years) (PR: 0.88, 95% CI: 0.76–0.99), in boys (PR: 0.87, 95% CI: 0.76–0.99), and in girls (PR: 0.84, 95% CI: 0.75–0.97). These results were not affected by additional adjustment for total energy intake or EI:BMR ratio.

### 3.4. Association of Dietary Patterns with Stunting

Adolescents who scored high on the nontraditional and breakfast-type (PR: 0.88, 95% CI: 0.81–0.95), Western (PR: 0.86, 95% CI: 0.80–0.92), and plant-based (PR: 0.85, 95% CI: 0.79–0.92) diets were less likely to be stunted (Table 6). With adjustment for EI:BMR ratio and stratifying by age and sex, the association between nontraditional and breakfast-type pattern was only significant for the age group 16–19 years and for boys. High scores on the Western pattern were inversely associated with stunting for the two age groups and for boys but not for girls. Finally, high scores on the plant-based pattern were significantly associated with lower prevalence of stunting for the 12–15 year age group, for boys, and for girls.

## 4. Discussion

In this study, four dietary patterns were identified, reflecting both Westernized/modern patterns (nontraditional and breakfast-type and Western) and traditional/transitioning patterns (plant-based and protein-rich). These findings are consistent with the nutritional transition in Mexico and with results reported in previous studies [7,29] and parallel the double burden of malnutrition. Kroker-Lobos et al. (2014) showed the coexistence of anemia, stunting, and overweight–obesity at individual, household, and national levels in Mexican children and women [30]. In addition, an association between mid-to-moderate household food insecurity and the co-occurrence of anemia and overweight–obesity in Mexican women has been reported [31]. Our results show that the double burden of malnutrition is also present at national and individual levels in Mexican adolescents. From 2006 to the present, the prevalence of overweight–obesity has increased, while the prevalence of anemia has decreased in Mexican adolescents. However, we do not have current dietary pattern data [12,13]. To our knowledge, this is the first study to link dietary patterns and indicators of the double burden of malnutrition at the individual level.

In the present study, the Western pattern was associated with malnutrition (overnutrition and undernutrition). Adolescents who scored high on the Western pattern were more likely to be overweight–obese and anemic. Similarly, high scores on the plant-based pattern were associated with a higher prevalence of overweight–obesity, but no association with anemia was observed. The Western and plant-based patterns had the highest correlation with total energy intake (0.36 and 0.47, *p* < 0.01) and sugar (0.38 and 0.23, *p* < 0.01). The correlation between the plant-based pattern and sugar can be explained by some of the food items that characterize this patter, such as nonindustrialized sweet drinks and sweet bakery. Both high energy and sugar intake have been associated with overweight and obesity [32,33,34,35]. Other studies conducted among adolescents from different countries have also shown that a Westernized dietary pattern increases the odds of being overweight or obese [7,36,37,38,39].

Anemia has been recognized as one of the three main causes of disability-adjusted life years among adolescents worldwide [40,41]. A systematic analysis of national surveys showed that the proportion of anemia associated with iron deficiency in women of reproductive age in Latin America was 59% [42]. Girls with a high intake of the Western pattern were more likely to be anemic. Although the Western pattern was positively correlated with iron intake (0.26, *p* < 0.01), adolescent girls have higher requirements of iron compared to boys [1]. Our results suggest that the content of iron in this pattern was not enough to cover the nutritional demands of adolescent girls. In addition, the bioavailability of iron in the Mexican diet has been shown to be low [11]. The associations between the nontraditional and breakfast-type pattern and lower prevalence of anemia in adolescents aged 12–15 years, both boys and girls, may be explained by nutrient interactions. This pattern was positively correlated with vitamin C (0.18, *p* < 0.01), which is an enhancer of iron absorption, and inversely correlated with fiber content (−0.30, *p* < 0.01), which inhibits iron absorption. However, anemia is not caused exclusively by iron deficiency. Other nutritional deficiencies, inflammatory status, and presence of parasites are factors that can also contribute to the development of anemia [42,43,44]. However, these data were not available in our study.

Stunting is an indicator of chronic undernutrition, which not only reflects poor nutrition but also the inadequacy of the environment to which a person has been exposed for a long period [45,46,47]. Therefore, the interpretation of a cross-sectional association between dietary patterns and stunting must be done with care. Stunting is mainly studied in children under five years old. However, adolescence is the second period of rapid linear growth after infancy, during which adolescents have a second opportunity to catch up on height [1,48]. Hence, we considered it important to study the association between dietary patterns and stunting in Mexican adolescents. We found that the prevalence of stunting was higher in the older age group of adolescents compared to the younger age group, which may be explained by a generation effect associated with the economic and nutritional transition. This would imply that adolescents in the older age group were more exposed to undernutrition during early childhood than in the younger group [49]. However, it may also reflect an early growth spurt with decelerating growth in later adolescence. The prevalence of stunting was lower in adolescents who scored high on the nontraditional and breakfast-type, Western, and plant-based patterns. Probably, adolescents in the fourth quartile of these three patterns have maintained a high caloric intake throughout their lives, which has allowed them to meet the caloric requirements to reach an optimal height for their age. In addition to energy intake, protein intake is essential for linear growth. In this study, the protein intake across all the quartiles of the four patterns fell into the recommended values for the age groups (34–52 g/day) [1]. Notwithstanding, the correlations between these patterns and energy (−0.04, 0.36, and 0.47, *p* < 0.01) as well as percentage of energy from protein (0.38, −0.05, −0.09, *p* < 0.01), iron (−0.06, 0.26, 0.27, *p* < 0.01), and zinc (0.11, 0.31, 0.38, *p* < 0.01) were inconsistent. Thus, drawing conclusions on the association between the actual dietary intake and prevalence of stunting is not straightforward.

This study has several limitations. Firstly, as with all cross-sectional studies, we could not determine causality between the adherence to a dietary pattern and nutritional status. Secondly, a seven-day FFQ was employed to collect dietary information. This FFQ provides an overview of what the participants ate the previous week and does not fully reflect habitual intake. Thirdly, as with all dietary intake assessments, this study was prone to measurement error and underreporting of energy-dense food. To overcome the last point, we used the EI:BMR ratio to adjust the analyses [50,51]. After adjusting for EI:BMR ratio, the direction of the associations between dietary patterns and overweight–obesity and stunting radically changed. Given the inverse correlation between BAZ and EI:BMR ratio (−0.13, *p* < 0.01), this may be due to underreporting of energy intake by subjects with high BAZ, which has also been suggested by a previous study in Mexican adolescents [52].

This study also has several strengths. We used PCA to derive dietary patterns, a methodology that involves several subjective decisions but provides a summary measure of eating habits, diet quality, and potential interactions between nutrients. Another strength is that our large sample size is representative at national, regional, and state levels. Therefore, our results are generalizable to the entire population of Mexican adolescents.

## 5. Conclusions

In conclusion, this assessment of the association between the main dietary patterns in Mexican adolescents in 2006 and indicators of malnutrition provides insight on how dietary patterns may increase the risk of one or more indicators of malnutrition. Although longitudinal studies are needed to assess causal associations, our results add to the existent literature that higher adherence to the Western pattern is associated with the double burden of malnutrition. The Western pattern was positively associated with overweight–obesity among all the adolescents and with anemia in girls. In the context of the double burden of malnutrition, dietary advice must consider malnutrition in all its forms. Given that adolescence is a crucial stage from a life-cycle perspective, efforts to improve nutritional habits during this stage can have a positive impact on the health of adolescents, their future adulthood, and for the next generation.

## Figures and Tables

**Figure 1 nutrients-11-02753-f001:**
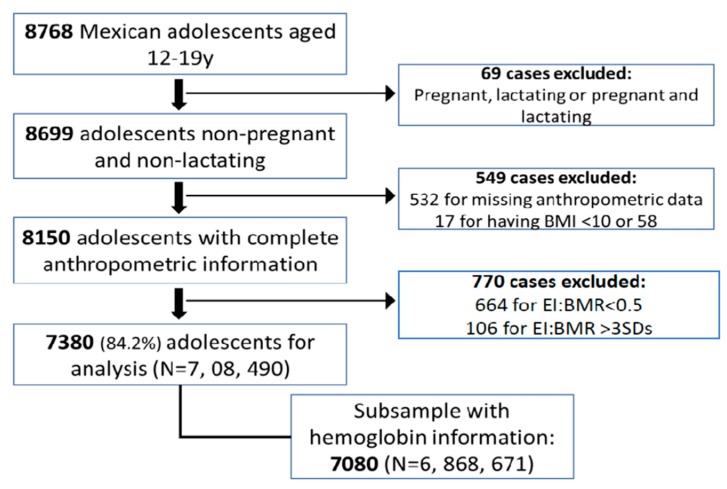
Selection process of study population. “N” represents the expanded sample. The sample of 7380 subjects represented 7,078,490 adolescents, while the subsample of 7080 subjects represented 6,868,671 adolescents. y = year; BMI = body mass index; EI:BMR = energy intake/basal metabolic rate ratio.

**Table 1 nutrients-11-02753-t001:** Sociodemographic characteristics and nutritional status of Mexican adolescents from ENSANUT-2006.

	Girls			Boys		
	12–15 Years (*n* = 2222)	16–19 Years (*n* = 1630)	*p*-Value	12–15 Years (*n* = 2377)	16–19 Years (*n* = 1377)	*p*-Value
Age Median (Range)	13.5 (12.0–15.0)	17.5 (16.0–19.0)		13.5 (12.0–15.0)	17.5 (16.0–19.0)	
Socioeconomic level (%)			0.38			0.08
Low	48.9	48.1		48.5	43.2	
Medium	32.9	31.2		31.0	31.8	
High	18.2	20.8		20.5	25.0	
Area of living			0.12			0.07
Urban	56.5	59.9		57.1	61.8	
Region			0.27			0.06
North	13.1	11.9		11.3	15.2	
Centre	45.5	48.7		48.6	46.3	
South	48.6	39.4		40.1	38.6	
Weight status (%)			0.86			0.015
Thinness	1.4	1.8		2.5	2.1	
Normal weight	66.6	67.2		67.2	71.8	
Overweight/Obese	31.9	31.1		30.3	26.2	
Stunting (%)	13.8	21.1	< 0.0001	14.3	19.2	0.012
Stunted subjects with overweight–obesity (%) ^£^	25.0	31.2		19.2	25.2	
Anemia (%) ^₰^	10.2	10.8	0.67	6.3	3.6	0.005
Anemic subjects with overweight–obesity (%) ꜜ	40.2	30.2		33.5	24.0	

Sample size: 7380; expansion factor: 7,078,490; ^₰^ sample size: 7080; expansion factor: 6,868,671. Anemia was estimated with hemoglobin concentration adjusted for geographic altitude; £ proportion of overweight–obesity among stunted subjects; ꜜ proportion of overweight–obesity among subjects with anemia.

**Table 2 nutrients-11-02753-t002:** Factor loading matrix for the four dietary patterns and their food groups.

Food Group	Average Intake (g/day)	Nontraditional and Breakfast-Type	Western	Plant-Based	Protein-Rich
Tortilla	194.5	−0.51			
Other cereals	29.6		0.33		
Breakfast cereals	5.0	0.63			
Maize-based food	50.2			0.36	
Fast-food	5.1		0.34		
Legumes	74.3				0.58
Fruit	186.2			0.54	
Vegetables	63.8			0.50	
Poultry and red meat	47.1				0.31
Fish and seafood	8.3				
Charcuterie	8.3		0.49		
Milk *	166.8	0.66			
Eggs	31.4				0.51
Fat	1.9		0.40		
Sweets	24.7	0.35			
Cookies	202.7				
Salty snacks	9.0		0.55		
Industrialized sweet drinks *	262.6		0.63		
Nonindustrialized sweet drinks *	256.2			0.53	
Dairy	33.5	0.37			
Sandwich	21.6	0.30	0.36		
Fried vegetarian dishes	6.6			0.36	
Sweet bakery	27.3			0.31	
Soup	66.3				0.50
Pasta and rice	23.7				0.57
Nuts and avocado	7.5			0.40	

Values from −0.30 through to 0.30 were excluded for simplicity in the interpretation. Average intake was calculated for each food item as the mean in grams per day. * (mL/d).

**Table 3 nutrients-11-02753-t003:** Correlation coefficients between dietary patterns and nutritional content.

	Nontraditional and Breakfast-Type	Western	Plant-Based	Protein-Rich
EI:BMR ratio	−0.04 *	0.29 *	0.48 *	0.29 *
Energy (kcal)	−0.04 *	0.36 *	0.47 *	0.33 *
Protein (% energy)	0.38 *	−0.05 *	−0.09 *	0.27 *
Fat (% energy)	0.32 *	0.43 *	−0.03 *	0.13 *
CHO’s (% energy)	−0.37 *	−0.36 *	0.05 *	−0.18 *
Fiber (g)	−0.30 *	−0.02 *	0.41 *	0.37 *
Sugar (g)	0.15 *	0.38 *	0.23 *	−0.01
Ca (mg)	0.15 *	0.06 *	0.38 *	0.20 *
Fe (mg)	−0.06 *	0.26 *	0.37 *	0.35 *
Zn (mg)	0.11 *	0.31 *	0.38 *	0.39 *
Vit C (mg)	0.18 *	−0.03 *	0.51 *	0.16 *
Vit A (IU)	0.25 *	−0.04 *	0.49 *	0.23 *
Vit B12 (mg)	0.40 *	0.33 *	0.26 *	0.23 *

* Significant at *p*-value 0.01.

**Table 4 nutrients-11-02753-t004:** Adjusted prevalence ratios of overweight and obesity for dietary patterns among Mexican adolescents.

Dietary Pattern	Total * (*n* = 7380)	12–15 years ^₰^ (*n* = 4451)	16–19 years ^₰^ (*n* = 2929)	Boys ^¥^ (*n* = 3610)	Girls ^¥^ (*n* = 3770)
Nontraditional and breakfast-type	1.03 (0.97–1.10)	0.98 (0.97–1.00)	0.91 (0.83–1.00)	1.01 (0.93–1.11)	1.05 (0.97–1.14)
Western	1.15 (1.08–1.21)	1.14 (1.06–1.22)	1.05 (0.95–1.16)	1.23 (1.12–1.34)	1.08 (0.99–1.18)
Plant-based	1.10 (1.03–1.17)	1.20 (1.11–1.30)	1.04 (0.92–1.17)	1.07 (0.97–1.18)	1.12 (1.04–1.22)
Protein-rich	1.02 (0.95–1.09)	0.97 (0.96–1.03)	1.05 (0.95–1.16)	1.06 (0.98–1.15)	0.98 (0.91–1.05)

All prevalence ratios were calculated per quartile of dietary pattern scores. * Model adjusted for sex, living area, socioeconomic status, region, age, and EI:BMR ratio. ^₰^ Model adjusted for sex, living area, socioeconomic status, region, and EI:BMR ratio. ^¥^ Model adjusted for living area, socioeconomic status, region, age, and EI:BMR ratio.

**Table 5 nutrients-11-02753-t005:** Prevalence ratios of anemia among Mexican adolescents across dietary patterns.

Dietary Pattern	Total * (*n* = 7080)	12–15 years ^₰^ (*n* = 4288)	16–19 years ^₰^ (*n* = 2792)	Boys ^¥^ (*n* = 3447)	Girls ^¥^ (*n* = 3633)
Nontraditional and breakfast-type	0.91 (0.81–1.01)	0.87 (0.76–0.99)	0.96 (0.80–1.16)	0.87 (0.76–0.99)	0.83 (0.73–0.96)
Western	1.11 (0.98–1.26)	1.11 (0.94–1.30)	1.13 (0.92–1.38)	1.11 (0.94–1.30)	1.24 (1.06–1.45)
Plant-based	0.99 (0.88–1.11)	1.04 (0.89–1.22)	0.87 (0.76–1.01)	1.04 (0.89–1.22)	0.96 (0.84–1.09)
Protein-rich	1.02 (0.92–1.12)	1.09 (0.95–1.25)	0.92 (0.79–1.06)	1.09 (0.95–1.25)	0.93 (0.83–1.05)

A subsample of 7080 subjects counted with hemoglobin data. Hemoglobin concentrations were adjusted for geographic altitude. All prevalence ratios were calculated per quartile of dietary pattern scores. * Model adjusted for sex, living area, socioeconomic status, region, age, and BMI-for-age Z-score (BAZ). ₰ Model adjusted for sex, living area, socioeconomic status, region, and BAZ. ¥ Model adjusted for living area, socioeconomic status, region, age, and BAZ.

**Table 6 nutrients-11-02753-t006:** Adjusted prevalence ratios of stunting among Mexican adolescents across dietary patterns.

Dietary Pattern	Total *	12–15 years ^₰^ (*n* = 4451)	16–19 years ^₰^ (*n* = 2929)	Boys ^¥^ (*n* = 3610)	Girls ^¥^ (*n* = 3770)
Nontraditional and breakfast-type	0.88 (0.81–0.95)	0.90 (0.80–1.00)	0.85 (0.77–0.95)	0.89 (0.81–0.97)	0.86 (0.77–1.03)
Western	0.86 (0.80–0.92)	0.88 (0.80–0.97)	0.85 (0.76–0.96)	0.81 (0.74–0.89)	0.93 (0.83–1.03)
Plant-based	0.85 (0.79–0.92)	0.80 (0.71–0.90)	0.90 (0.80–1.02)	0.80 (0.71–0.90)	0.80 (0.72–0.89)
Protein-rich	0.96 (0.90–1.02)	0.92 (0.84–1.01)	1.00 (0.91–1.10)	0.92 (0.84–1.01)	0.98 (0.90–1.07)

All prevalence ratios were calculated per quartile of dietary pattern scores. * Model adjusted for sex, living area, socioeconomic status, region, age, and EI:BMR ratio. ^₰^ Model adjusted for sex, living area, socioeconomic status, region, and EI:BMR ratio. ^¥^ Model adjusted for living area, socioeconomic status, region, age, and EI:BMR ratio.

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
