# Peer review of "Dietary Patterns and the Double Burden of Malnutrition in Mexican Adolescents: Results from ENSANUT-2006"

_nutrients, 2019, doi:10.3390/nu11112753_

Round 1

Reviewer 1 Report

The present paper seems to be an important new piece of literature describing the relationships between dietary patterns and health outcomes in Mexican adolescents using nationally representative data. The authors are to be commended for their extremely thorough analysis approach and important subsample analysis. I appreciated availability of supplemental data as well. Overall, I suggest minor-moderate edits in the following sections to improve the manuscript.

The introduction:

In line 38, it is not clear why the term "in turn" was used since the sentence marks the beginning of a new paragraph that is not logically linked to the previous paragraph/sentence. I would suggest to delete "in turn."  In line 46 of the introduction and line 233 of the discussion (and other places of the discussion), the term "double burden of malnutrition" is used. Is this a commonly accepted term or a term coined by the authors? This paragraph is rather hard to follow. It would help tp replace non-communicable diseases with "diet-related chronic conditions." Line 53: replace "up till now" with "up until now." 

Methods:

Line 138: make boy plural (boys).

Results:

Table 1: Replace "Thinness" with "Underweight." Table 1: For the categories under stunting and anemia in boys, are the rows misaligned? Dietary patterns: it is not clear why the non-traditional and breakfast/lunch-type pattern carries the term "lunch." Sandwiches are not necessarily lunch foods and there is no indication of any other typical lunch food in this pattern. I would suggest reconsidering the label of this DP to exclude the term "lunch." I would have liked to see more discussion on the fact that the main results highly depended on controlling for EI:BMR ratio (table 4 vs. table 3). It appears that when the ratio is not controlled for, the protein-rich diet is negatively associated with overweight/obesity in younger adolescents and in girls.

Discussion:

Why is the plant-based diet associated with high sugar intake? What foods are high in sugar and energy that would explain the association with overweight/obesity? Please elaborate on this finding in the discussion. The discussion contains adequate information on expected findings, but not enough discussion of unexpected findings, such as the association between the plant-based diet and overweight/obesity and the disappearance of this finding when controlling for EI:BMR ratio. This is probably the most important discussion point of this paper and needs to be addressed (mentioning it in the limitations paragraph is not sufficient).

Author Response

Dear Reviewer,

We appreciated your comments. We feel that your feedback has help us to improve the manuscript. Please, see attached the rebuttal letter, were we responded to your point one by one. You also can see the attached word file with the corresponding corrections. 

Kind Regards,

Arli Zarate

Reviewer 2 Report

This manuscript reports the results of a PCA of dietary patterns in Mexican adolescents and their association with overweight and undernutrition (stunting and anemia). The manuscript is clearly written and concise, and the statistical methods are appropriate. I have some minor suggestions. 

Line 34 - reference needed for this sentence. 

line 38 - the first sentence of this paragraph is not well integrated with the remainder of the paragraph. I'd suggest restructuring the introduction to ensure each paragraph deals with a specific topic (e.g., the bulk of the information in this paragraph could be discussed in the first paragraph when making a case for the importance of considering nutrition and eating patterns in adolescence).

The introduction should explicate what the authors use as their dependent measures of 'overweight' and 'undernutrition'. Specifically, for the latter, define why anemia was chosen as an outcome measure and not another indicator of undernutrition? Potentially there are other conditions / indicators that could have been considered? 

Line 44 - 'Western' dietary pattern is mentioned but not defined here. 

The introduction is very brief. What is the 'dietary pattern' approach introduced at line 57? If there is no evidence in Mexican adolescents, can the authors review evidence from studies taking a dietary pattern approach in other populations? A more comprehensive background literature review is needed.

Grammar line 53 - 'up till' replace with 'until'.

Line 63 - representative of whom? Specify what population /age group etc.

The authors comprehensively report and mostly justify the selection of their sample, but should specify the rationale behind the plausible BMI cut off values stated at line 73, and the rationale behind considering to age groups 12-15 and 16-19.

Further, briefly stating what is meant by 'expanded pool' of adolescents would be useful to enhance clarity (eg line 76).

I appreciated the inclusion of Figure 1, but suggest slight revision as it is a little difficult to read (it appears stretched and the text very small). Given the large amount of white space, the authors should be able to restructure the figure to enhance clarity without sacrificing page space. Arrow heads should be added to indicate the logical flow in vertical and horizontal arrows (e.g., between text box beginning '8768' down vertically to text box beginning '8699', and eg between that vertical line and text box beginning '69').

Additional detail on how the 30 food groupings were decided upon (mentioned line 101) is needed. Is this an accepted approach to classifying food for this population?

In Supplemental Table 1, the group 'supplements' is missing defining examples.

The authors should be commended for preparing and clearly reporting a comprehensive and well-justified analysis strategy and for reporting the results of unadjusted as well as adjusted regression analyses.

Is it not surprising that the 'protein-rich' group was not associated with lower prevalence of stunting given the importance of protein for growth?  

Elaboration on the concept of 'nutritional transition' in Mexico (mentioned at line 232) would be useful in the introduction and in the discussion. 

It's not clear to me what the authors mean by 'coexistence of... at individual, household, and national level...' at line 234 and later at line 238 ('is also present at national and individual level'). Clarification is needed.

Grammar line 243: replace 'showed to be' with 'was' and specify 'in the present study'.

Line 308 - longitudinal studies can still not establish causation - experimental work is needed for cause-effect conclusions (which is not feasible for population-based nutrition studies).

Line 310 - the authors should rephrase to avoid implying a cause-effect relationship ('the western pattern increases the likelihood' - the term 'increases' implies the dietary pattern is a causal factor).

Author Response

Dear Reviewer,

We appreciated your comments. We feel that your feedback has help us to improve the manuscript. Please, see attached the rebuttal letter, were we responded to your points one by one. You also can see the attached word file with the corresponding corrections. 

Kind Regards,

Arli Zarate
